# General-Purpose Deep Learning Detection and Segmentation Models for Images from a Lidar-Based Camera Sensor

**DOI:** 10.3390/s23062936

**Published:** 2023-03-08

**Authors:** Xianjia Yu, Sahar Salimpour, Jorge Peña Queralta, Tomi Westerlund

**Affiliations:** Turku Intelligent Embedded and Robotic Systems Laboratory, Faculty of Technology, University of Turku, 20500 Turku, Finland

**Keywords:** deep learning, object detection, instance segmentation, semantic segmentation, lidar, lidar-based perception

## Abstract

Over the last decade, robotic perception algorithms have significantly benefited from the rapid advances in deep learning (DL). Indeed, a significant amount of the autonomy stack of different commercial and research platforms relies on DL for situational awareness, especially vision sensors. This work explored the potential of general-purpose DL perception algorithms, specifically detection and segmentation neural networks, for processing image-like outputs of advanced lidar sensors. Rather than processing the three-dimensional point cloud data, this is, to the best of our knowledge, the first work to focus on low-resolution images with a 360° field of view obtained with lidar sensors by encoding either depth, reflectivity, or near-infrared light in the image pixels. We showed that with adequate preprocessing, general-purpose DL models can process these images, opening the door to their usage in environmental conditions where vision sensors present inherent limitations. We provided both a qualitative and quantitative analysis of the performance of a variety of neural network architectures. We believe that using DL models built for visual cameras offers significant advantages due to their much wider availability and maturity compared to point cloud-based perception.

## 1. Introduction

Autonomous mobile robots and self-driving cars use a variety of sensors for ensuring a high level of situational awareness [1]. For instance, the Autoware project, representing state-of-the-art autonomous cars, relies on 3D lidars for key perception components [2]. Multiple aerial robotic solutions also utilize lidars for autonomous flight in complex environments [3]. Some of the critical characteristics of lidar that motivates its adoption across application fields include its long range and the accuracy of the geometric data it outputs.

Lidar point cloud data feature 360°, three-dimensional, high spatial resolution data, but often have a limited vertical field of view. Advanced sensors have vertical resolutions that typically range from 30° to 90° [4]. As lidar measures the time of flight of a laser signal to objects in the environment, it is not influenced by changes in light, such as darkness and daylight. In several studies, lidar point cloud data and image data have been used together in a variety of computer vision tasks, such as 3D object detection [5,6]. However, while lidar odometry, localization, and mapping are at the pinnacle of autonomous technology [7], the processing of point cloud data for object detection or semantic scene segmentation is not as mature as the algorithms and machine learning (ML) approaches for vision sensors [8,9].

Deep learning (DL) has revolutionized computer vision over the last decade within the robotics field, from advanced perception [10] to novel end-to-end control architectures based on deep reinforcement learning [11], including odometry and localization [9]. For this work, we were particularly interested in DL models for object detection and instance segmentation, both of which are cornerstones to embedding intelligence into autonomous robots and enabling high degrees of situational awareness [12]. Even though most of the work in DL-based perception has focused on images and vision sensors, DL applications to lidar data include voxel-based object detection or point cloud segmentation [13]. The literature also includes multiple examples of lidar and camera fusion for producing colored point clouds or more robust behavior, e.g., when segmenting roads in self-driving cars [14]. These works, however, focus on point cloud lidar data [13], while we explore the potential to leverage them as camera-like sensors. This potential has only recently been identified [15], and the existing literature lacks a more in-depth analysis of the potential of images captured from lidar sensors. A sample of the data used in this work is shown in Figure 1. We refer the reader to existing dataset papers with this type of data for a more in-depth characterization of the different types of images that the Ouster lidar sensors can generate [16].

Although lidar sensors currently cost more than passive visual sensors, lidar sensors are inherently more robust, withstanding adverse weather conditions and low-visibility environments. They are also a standard part of most of today’s self-driving autonomy stacks. Therefore, it comes at no extra cost to leverage their vision-like capabilities, in addition to processing the three-dimensional point cloud data.

The main contribution of this work is the analysis of the performance of a variety of DL-based visual perception models in lidar camera data. We assessed the viability of applying object detection and instance segmentation models to low-resolution, 360° images from two different Ouster lidar sensors with different fields of view and range.On the object detection side, we utilized both one-stage detectors (YOLOv5 and YOLOx) and two-stage detectors (the Faster R-CNN and the Mask R-CNN). For semantic instance segmentation, we studied the performance of the HRNet, PointRend and the Mask R-CNN.

The remainder of this document is structured as follows. Section 2 contains an overview of the literature on DL perception, lidar-based object detection and segmentation, and the fusion of vision and lidar sensors. Section 3 covers the hardware and software methods utilized in our work. In Section 4, we report experimental results, and we discuss the potential of this type of sensor data in Section 5. Finally, Section 6 concludes the work and outlines future research directions.

## 2. Related Work

Literature on the processing of low-resolution lidar-based images is scarce. In [17], Ouster’s CEO introduced the technology, showcasing the performance of car and road segmentation using a retrained DL model with a video. The author also commented on the potential for using these data as input to a pretrained network from DeTone et al.’s SuperPoint project for odometry estimations. However, in both cases, the code was not available, and the quantitative results were not shown. Minimal research has been carried out in this direction to the best of our knowledge. In [15], Tsiourva et al. analyzed the potential of the same Ouster lidar sensors that we studied for saliency detection. This work already demonstrated more consistent performance and data quality in adverse environments (e.g., rainy weather). We further analyzed DL-based perception performance beyond essential computer vision preprocessing, such as saliency detection.

A relevant recent work in the literature is [16], where the authors presented a novel dataset of lidar-generated images with the same lidar-as-a-camera sensor that we used in this paper. The work in [16] showed the potential of these images, as they remained almost invariant across seasonal changes and environmental conditions. For example, unpaved roads could be perceived in very similar ways in summer weather, snow cover, or light rain. Therefore, there was a clear advantage in using these images over using standard RGB images or even images from infrared cameras, despite the limited vertical resolution.

Through the rest of this section, we review the current research directions and the state-of-the-art in lidar-based perception and fusion with cameras, DL-based object detection and segmentation, and the fusion of lidar and camera data.

### 2.1. Lidar-Based Perception

Lidar data provide an accurate and reliable depth and geometric information, and they are a crucial component of various kinds of perception tasks, such as 3D mapping, localization, and object detection [7].

There have been many studies carried out on detection and localization tasks using lidar point cloud data [18,19]. In most cases, however, current techniques are based on a fusion of both camera and lidar data [20,21,22]. In [14] different fusion methods were applied to detect roads with lidar and camera data. In addition, several studies have utilized lidar and camera data to detect pedestrians and vehicles with self-driving systems [23,24].

Despite the rapid advances in recent years that the works above show, processing 3D lidar point cloud data is still significantly more expensive in terms of resources than processing images [25]. Additionally, the methods are usually purpose-built and specific to use cases or application scenarios, limiting generalizability to, for example, detecting different types of objects.

### 2.2. Deep Learning-Based Object Detection

Object detection has been among the most trivial tasks in computer vision applications. This task has been extensively explored in a wide range of technological advances in recent years, including autonomous driving, identity detection, medical applications, and robotics. In most state-of-the-art object-detection methods, deep learning neural network models are used as the backbone to extract features and classify objects and identify their locations [10,12].

The most popular types of detectors are the YOLO [26] (You Only Look Once) algorithm-based detector and various versions of it [27,28], RetinaNet [29], the SSD [30] (Single Shot MultiBox Detector), the R-CNN [31] (Region-CNN) and its extensions, and the Mask R-CNN [32].

A representative example appears in [19], where the authors proposed a 3D fully convolutional network based on DenseBox for 3D detection and localization of vehicles from lidar point cloud data. As described in [33], RGB camera data and lidar point cloud data were combined to enhance the object detection performance in real time by using a weighted-mean YOLO algorithm. In other approaches, point cloud data were converted into bird’s eye view images and then fused with front-facing camera images using multiview 3D networks to predict 3D bounding boxes [34,35].

In summary, there has been an exponential increase in research in deep learning and computer vision approaches to object detection. The field is significantly more mature than the field of lidar data processing, but there is a gap in the literature in terms of the study of the applicability of these methods to other types of images generated with different sensors. In this work, we aimed to study these potential applications.

### 2.3. Deep Learning-Based Instance and Semantic Segmentation

In [36], a dual-modal instance segmentation deep neural network based on the architectures of the RetinaNet network and the Mask R-CNN was developed for object segmentation using the RGB and Lidar pixel-level images. The authors of [37] transformed the 3D lidar point clouds into 2D grid representations by applying a spherical projection. Then, the SqueezeSeg model derived from SqueezeNet was developed for the semantic segmentation of the obtained range images. Alternatively, in [38], the authors proposed a transfer learning model based on MobileNetv2 for the semantic segmentation of a birds-eye-view representation of the 3D point cloud data.

Similar to object detection research, there are more mature methods in terms of computer vision for semantic segmentation, compared to lidar-based segmentation. Bringing the benefits of image-based methods to lidar sensors has the potential to increase the degree of situational awareness achieved across environmental conditions where passive vision sensors do no perform as well as lidar sensors.

## 3. Methodology

This section covers the hardware and methods utilized in our study. We describe the sensors utilized for data acquisition as well as the different DL model architectures.

### 3.1. Hardware

The equipment for data acquisition consisted of two spinning lidar sensors: the Ouster OS1-64 and the Ouster OS0-128. Table 1 shows the key specifications of these lidar sensors, including the resolution of the images that they generated. It is worth noting that the vertical resolution of the images matched the number of channels in the lidar sensor.

Figure 2 depicts the data collection platform that was mounted on different mobile platforms. The two lidar sensors were installed on the sides, while an Intel RealSense L515 lidar camera captured RGB images.

### 3.2. Data Acquisition

We gathered data in various settings, including indoors and outdoors, and during both day and night. For this initial assessment of the performance of the DL models on images generated by the lidar sensors, we concentrated on a selection of object categories. These categories were chosen based on the typical needs of autonomous systems, as well as on objects that appeared more often in the collected data. Outdoors, we analyzed the detection of cars, bicycles, and persons. Indoors, we analyzed the detection of persons and chairs. Table 2 shows the number of object instances in the collected data. Samples of the data generated by the sensors are shown in Figure 1. In these examples, the resolution of a lidar-generated image was 2048×128 with a $360^{\circ }$ field of view of the surrounding scene, while an RGB image from the L515 had a resolution of 1920×1080.

While we did not study invariability of object detection or segmentation of the same class of objects across different environments (e.g., indoors and outdoors) or environmental conditions (e.g., light rain or fog, day or night), we could assume, based on the works in the literature [16], that the data characteristics did not change significantly. Indeed, one of the key benefits of lidar-generated images is that they are not affected by environmental conditions. Therefore, a person is detected in an almost invariant manner both indoors and outdoors, as long as it is at the same distance and relative position to the sensor. The same applies to lidar images generated in daylight or at nighttime.

### 3.3. Data Preprocessing

One of the main drawbacks of lidar-generated images is the low vertical resolution, which is only up to 128 pixels in the highest-performance sensors. Our early experiments showed low performance in the different detection and segmentation models due to the high distortion in the untraditional image ratio. To address this issue with data preprocessing, we performed two main steps: denoising and interpolation using the OpenCV libraries with Python. We considered different denoising and interpolation approaches and repeatedly ran object detection and segmentation on a set of test images. In our experiments, we applied a box filter to denoise the images and linear interpolation methods to properly resize the images to the dimension of 1000×300. Figure 3 shows the original signal image in Figure 3a and the one after the preprocessing in Figure 3b.

### 3.4. Object Detection Approaches

Over the last decade, deep neural network models have achieved significant advances in computer vision, especially object detection. Object detection, which includes both object recognition and localization, is generally divided into two types: one-stage and two-stage detection [39]. In this study, some of the most commonly used models from both frameworks were utilized for object detection.

#### 3.4.1. Two-Stage Object Detection

A two-stage detector divides the detection process into region proposal and classification phases. At the region proposal phase, several object candidates are proposed as regions of interest (RoIs), which are classified and localized in the second phase. Object localization and detection are typically more accurate in models with a two-stage architecture than in others. Two popular two-stage detectors were used in this study: the Faster R-CNN [40] and the Mask R-CNN [32]. These models were implemented based on Pytorch, and ResNet-50 was used as the pretrained backbone for object detection.

#### 3.4.2. One-Stage Object Detection

In contrast to two-stage models, one-stage detectors utilize a single feed-forward, fully-convolutional network for object feature extraction, bounding-box regression, and classification. In the one-stage approach, feature maps are detected and classified simultaneously. In addition to their excellent accuracy, the one-stage detector models are popular in real-time applications due to their high detection speed. One of the first widely adopted one-stage detectors in the deep learning field was YOLO, which was introduced in [26]. Two variations of the YOLO model were applied in this study: YOLOx [41] and YOLOv5 [42]. In the YOLOx toolset, there are different types of networks, including the YOLOx-s, YOLOx-m, YOLOx-l, and YOLOx-x models. We used the YOLOx-m model in this paper due to its high detection speed and performance.

### 3.5. Image Segmentation Approaches

Object segmentation is the process of assigning each pixel value of an image to a specific class, and it is generally divided into two types: semantic segmentation and instance segmentation. The semantic segmentation method considers objects that belong to the same class as a single group [43], while the instance segmentation method combines semantic segmentation and object detection approaches and identifies multiple objects of a single class as distinct instances [44].

For semantic segmentation, HRNet + OCR + SegFix (a high-resolution network), which placed first in the Cityscapes competition at ECCV 2020, was used [45]. HRNet + OCR + SegFix is the integration of HRNet, OCR, and SegFix to provide a powerful tool for the precise localization of text or objects in images that require high-resolution feature extraction. HRNet is a DL architecture designed for high-resolution images that capture fine-grained details and global context through a parallel multi-resolution pyramid structure. OCR is an optical character recognition technology that allows computers to recognize and interpret text in images. SegFix is a postprocessing technique for image segmentation that corrects errors by using context from neighboring pixels.

Additionally, Pointrend [46] and the Mask R-CNN, both with ResNet-50 as their backbone, were employed for instance segmentation. Particularly, PointRend is a cutting-edge technique for instance segmentation, which predicts point-wise predictions for each pixel in an image and selectively refines them based on context using a context-adaptive CNN. This selective refinement approach achieves state-of-the-art results with fewer computational resources than traditional instance segmentation techniques. PointRend is flexible and easily integrated into existing pipelines, making it a popular technique in computer vision. It has demonstrated impressive results on various datasets.

## 4. Experimental Results

Through this section, we cover the results of applying the different object detection and instance segmentation models to the data gathered in the different environments. We collected and manually annotated the lidar-generated signal images. We used RGB images from a separate camera and lidar point cloud data to validate the annotations through visual observation.

### 4.1. Detection Results

The first part of the analysis delved into the performance of the different objectors.

Table 3 shows the proportion of objects successfully detected by the Faster R-CNN, the Mask R-CNN, YOLOv5, and YOLOx. Among them, YOLOx had a higher proportion of detected objects indoors and outdoors. It is worth noting that all four models were able to detect over 80% of persons indoors and over 80% of cars outdoors. In general, the performance of all the models was good enough to consider the adoption of this type of object detection in systems where lidar sensors are already present.

For more specific metrics, Table 4 shows the precision and recall of the detectors. YOLOx had the most robust overall performance of the four different tested models.

Some other categories, including stop signs, handbags, and fire hydrants, were considered in our initial evaluation. However, they are not listed in Table 3 and Table 4 as we focused on better analyzing a specific subset. In general, we have observed that both YOLOv5 and YOLOx could achieve comparable accuracy in these other classes as well.

Figure 4 shows a sample of detection examples from the following methods for both indoor and outdoor scenes: YOLOv5, Figure 4b; YOLOx, Figure 4a,c; the FasterR-CNN, Figure 4d; and the Mask R-CNN, Figure 4e.

In our experiments, single-stage object detectors outperformed the two-stage methods. The literature in the area points to the better overall performance of two-stage models. However, for the data studied in this paper, this did not hold. In any case, the limited amount of data for our tests was not enough to conclude that single-stage detectors are always better for lidar-generated data.

### 4.2. Segmentation Results

Regarding the performance of instance segmentation models, Figure 5 shows examples of HRNet semantic segmentation in both indoor and outdoor scenes. Figure 6 also shows examples of the instance segmentation results with PointRend and the Mask R-CNN. In this case, the analysis was qualitative, and further results are available in the project’s repository https://github.com/TIERS/lidar-as-a-camera (accessed on 3 March 2023). Nonetheless, our tests showed good performances for the most typical object classes based on analyzing a broad series of images.

### 4.3. Real-Time Performance Evaluation

We evaluated the real-time performance of multiple representatives from the above approaches including YOLOv5, the Faster R-CNN from detection tasks, and PointRend from the segmentation tasks. The computing platform utilized was an Nvidia GeForce RTX 3080 GPU with 16GB GDDR6 VRAM. The YOLOv5 with YOLOv5s model had an average inference frequency of 24 HZ. The Faster R-CNN with ResNet50 FPN model averaged to 15 HZ. Additionally, the PointRend with ResNet50 had the backbone average of 15 HZ.

## 5. Discussion

The results from our experiments demonstrated the potential use of lidar sensors as camera sensors with out-of-the-box, state-of-the-art DL-based perception models. Indeed, this type of data processing pipeline came at no extra cost for autonomous systems where lidar sensors are already present.

The positive performance of both object detectors and semantic instance segmentation networks opens the door to the broader use of these sensors for perception beyond the analysis of the three-dimensional point clouds. Our work extended the results from previous experiments in [15] that already showed the potential of lidar sensors as cameras, especially in adverse weather conditions and environments where passive vision sensors present inherent limitations.

Among the applications of the use of lidar sensors as cameras in autonomous robotic solutions is their potential for odometry estimations and aiding both lidar and visual odometry with an intrinsically multimodal data source. This was loosely explored in [17], but further research is needed to quantify the performance of such an approach.

One aspect that also requires further study is utilizing the different images generated by the lidar sensors. Through this work, we decided to focus on one of the three types of images provided by the lidar sensors, namely the signal image. In addition to this, the sensors also provided depth, near-infrared and reflectivity images. These, however, did not perform as well with out-of-the-box DL models without further preprocessing, as we illustrated with a sample in Figure 4a and Figure 7. One option to be further explored in the future is the combination of these images as multiple channels of a single image.

## 6. Conclusions and Future Work

In this work, we presented an analysis of the performance of different object detection and semantic segmentation DL models on images generated by lidar sensors. We collected data with two different lidar sensors indoors and outdoors and in both daylight and night scenes. Our experiments showed that state-of-the-art DL models could process this type of data with a promising performance by interpolating the low-resolution images to adequate resolutions. Object segmentation results were particularly optimistic, therefore paving the path for further usage of lidar sensors beyond the current algorithms focused on odometry, localization, mapping, and object detection from geometric methods. The main limitation of the current analysis is perhaps the lack of retraining for the models with larger datasets of lidar-generated images, owing to the lack of such annotated datasets.

In future work, we will explore a wider variety of preprocessing techniques and study the performance benefits of retraining some of the studied network architectures with data from the lidar camera sensors.

## Figures and Tables

**Figure 1 sensors-23-02936-f001:**
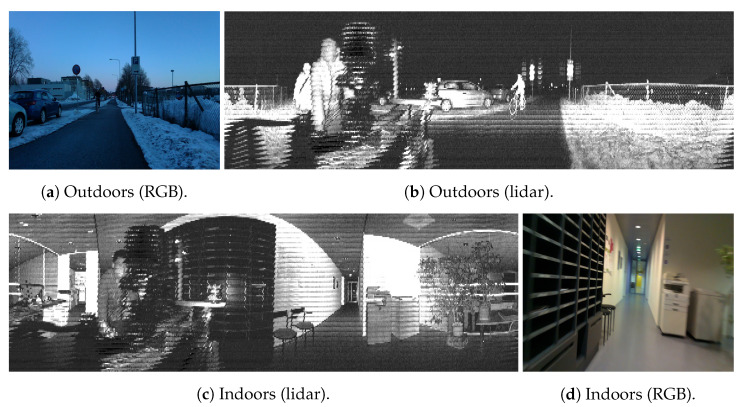
Samples of images utilized in this work. The outdoor sample includes a bicycle that is seen in both the RGB and lidar data, as well as several cars. In both the indoor and outdoor images, a person behind the sensors appeared in the 360° lidar image but not in the RGB frame.

**Figure 2 sensors-23-02936-f002:**
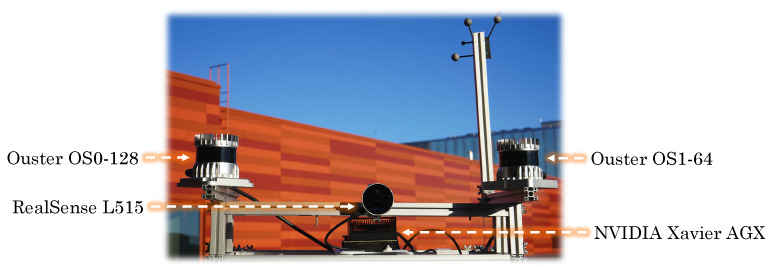
Equipment utilized for data acquisition.

**Figure 3 sensors-23-02936-f003:**
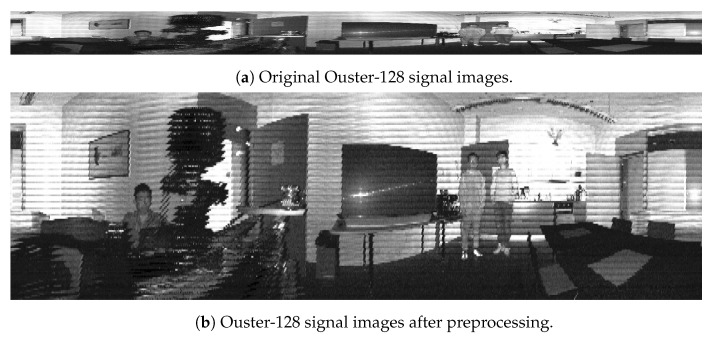
Ouster signal images before and after preprocessing.

**Figure 4 sensors-23-02936-f004:**
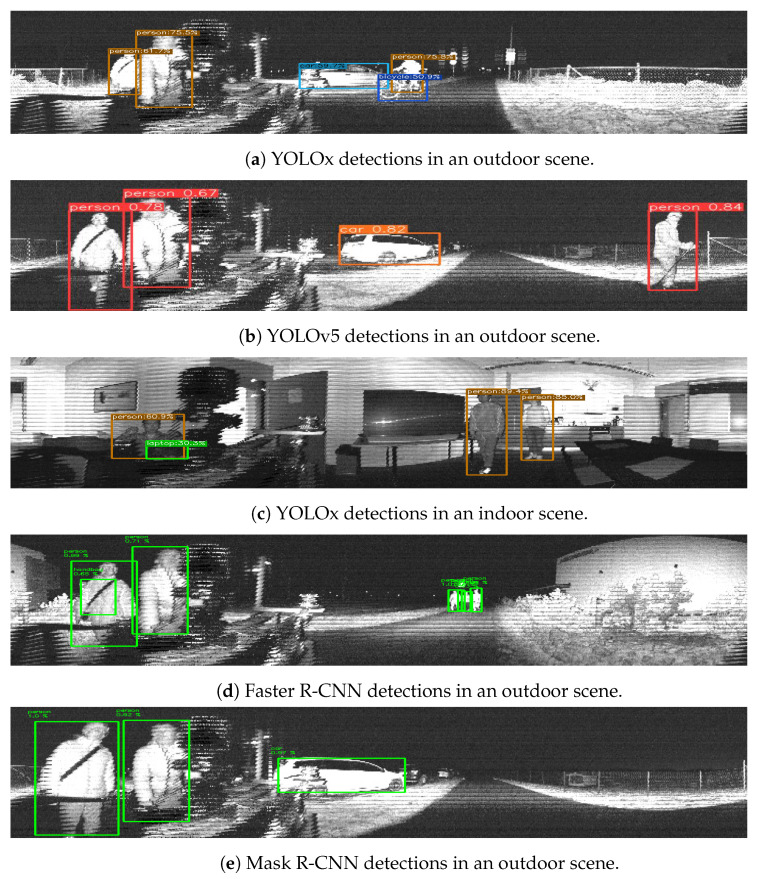
Detection examples in indoor and outdoor scenarios.

**Figure 5 sensors-23-02936-f005:**
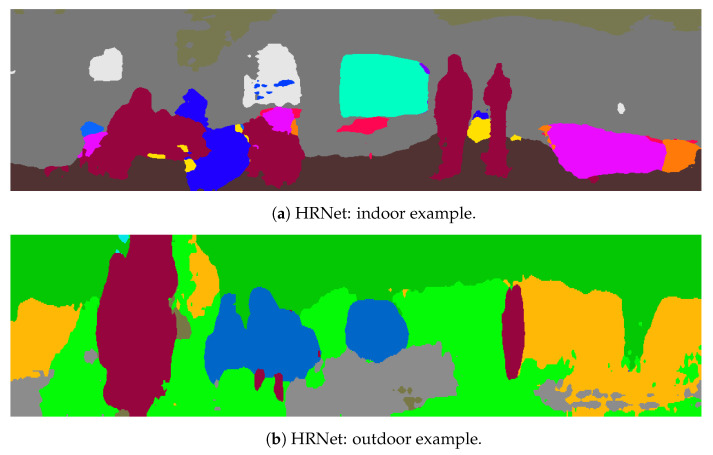
Indoor and outdoor semantic segmentation examples based on HRNet.

**Figure 6 sensors-23-02936-f006:**
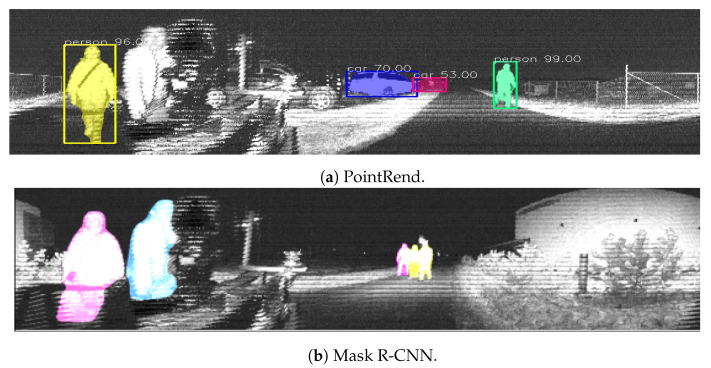
Indoor and outdoor instance segmentation examples based on PointRend and the Mask R-CNN.

**Figure 7 sensors-23-02936-f007:**
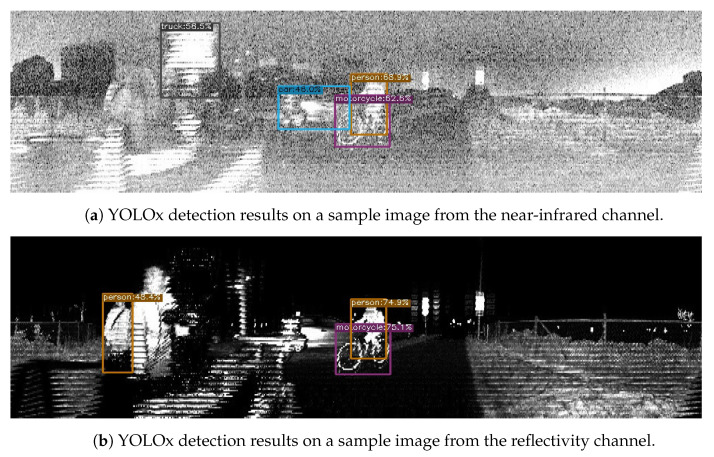
YOLOx detection based on images from other channels of Ouster lidar sensors.

**Table 1 sensors-23-02936-t001:** Specifications.

	Channels	FoV	Range	Frequency	Image Resolution
**Ouster OS1-64**	64	360° × 45°	120 m	10 Hz	2048 × 128
**Ouster OS0-128**	128	360° × 90°	50 m	10 Hz	2048 × 64
**RealSense L515**	N/A	70° × 55°	9 m	30 Hz	1920 × 1080

**Table 2 sensors-23-02936-t002:** Instances of the different objects in the analyzed dataset.

	Indoors		Outdoors		
	Person	Chair	Person	Car	Bike
Instances	43	42	103	37	14

**Table 3 sensors-23-02936-t003:** Proportion of objects successfully detected by each of the models studied in this work. This metric did not include false negatives or false positives.

		Faster R-CNN	Mask R-CNN	YOLOv5	YOLOx
**Indoors**	Person	0.837	0.837	0.924	**0.953**
	Chair	0.357	0.333	0.398	**0.515**
**Outdoors**	Person	0.524	0.485	0.630	**0.633**
	Car	0.865	0.811	**0.893**	0.866
	Bike	0.357	**0.643**	0.143	**0.571**

**Table 4 sensors-23-02936-t004:** Detection accuracy of multiple representative object detection networks in various scenarios.

		Faster R-CNN		Mask R-CNN		YOLOv5		YOLOx	
		Precision	Recall	Precision	Recall	Precision	Recall	Precision	Recall
**In**	Person	0.72	0.837	0.95	0.905	0.976	0.930	**1.0**	**0.953**
	Chair	1.0	0.115	0.57	**0.826**	1.0	0.115	**1.0**	0.315
**Out**	Person	0.912	0.505	0.957	0.464	0.872	0.854	0.969	0.653
	Car	0.943	0.688	0.712	0.627	0.919	0.829	0.825	0.618
	Bike	0.357	1.00	0.643	1.00	0.143	1.00	0.571	1.00

## Data Availability

Not applicable.

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
