# Peer review of "General-Purpose Deep Learning Detection and Segmentation Models for Images from a Lidar-Based Camera Sensor"

_sensors, 2023, doi:10.3390/s23062936_

Round 1
Reviewer 1 Report
This paper reports on a novel detection and segmentation framework using low-resolution images with 360°field of view obtained with lidar sensors. This is a very interesting and useful work. Overall, this work is complete, but improvements are still necessary.
Detailed comments:
1. As stated by the author in Introduction, multi-image sensors play very important role for the perception of autonomous mobile robots and self-driving cars, e.g., visible, depth and infrared images. The author needs to briefly discuss the advantages and disadvantages of each perceived image, e.g., 10.1109/TMECH.2022.3215909, 10.1109/LRA.2022.3168335. In other words, what is the author's motivation to only study lidar images rather than multi-image fusion sensors.
2. The running time of each algorithm has not been analyzed in detail to meet the real-time requirements.
3. How are the presented data sets labeled? For the radar images, it's hard to distinguish what type of object is it. The authors need to explain how to deal with this problem.
Author Response
************** Responses to Reviewer 1 ******************
1.1 Comment:
This paper reports on a novel detection and segmentation framework using low-resolution images with 360°field of view obtained with lidar sensors. This is a very interesting and useful work. Overall, this work is complete, but improvements are still necessary.
1.1 Response:
Thanks for your complimentary comments. We have addressed the issues mentioned below to improve the quality of the manuscript. We have restructured some of the sections and carried out some new experiments to explicitly discuss the issues concerned.
-------------------------------------------------------------------------------------------------------------------
1.2 Comment:
As stated by the author in Introduction, multi-image sensors play a very important role for the perception of autonomous mobile robots and self-driving cars, e.g., visible, depth and infrared images. The author needs to briefly discuss the advantages and disadvantages of each perceived image, e.g., 10.1109/TMECH.2022.3215909, 10.1109/LRA.2022.3168335. In other words, what is the author's motivation to only study lidar images rather than multi-image fusion sensors.
1.2 Response:
Many thanks for your feedback, we understand that the motivation was not clear enough in the manuscript. We have rephrased parts of the Introduction section to better explain the motivation between studying lidar-generated images. In short, the motivation is that lidars are already used in multiple types of autonomous mobile robots or vehicles, and we can add intelligence to the system without additional sensors (e.g., multiple cameras). Additionally, the lidar-generated images are, in general, more robust to variable weather conditions, particularly in low-light, as we show in the paper.
1.2 Action:
We have rephrased the motivation across multiple paragraphs in the introduction and conclusion sections. We also added a new related work, “LiDAR-as-Camera for End-to-End Driving”, that shows the clear advantages of lidar images for multi-season and variable weather datasets, including snow and rain. The lidar-generated images remain more invariant than standard vision sensors or RGB cameras.
-----------------------------------------------------------------------------------------------------------------
1.3 Comment:
The running time of each algorithm has not been analyzed in detail to meet the real-time requirements.
1.3 Response:
Several reviewers have pointed out the lack of a runtime analysis. We thank you for pointing this out, we have added new results to the paper.
1.3 Action:
We have new evaluation results and a new subsection referring to subsection 4.3 discussing multiple detection and segmentation inference frequencies. Specifically, “we evaluated the real-time performance of multiple representatives from the above approaches including YOLOv5, Faster R-CNN from detection tasks, and PointRend from the segmentation tasks. The computing platform utilized is an Nvidia GeForce RTX 3080 GPU with 16GB GDDR6 VRAM. YOLOv5 with YOLOv5s model has an average inference frequency of 24 HZ. Faster R-CNN with ResNet50 FPN model averages to 15 HZ. Additionally, the PointRend with ResNet50 as the backbone averages 15 HZ”.
----------------------------------------------------------------------------------------------------------------------
1.4 Comment:
How are the presented data sets labeled? For the radar images, it's hard to distinguish what type of object is it. The authors need to explain how to deal with this problem.
1.4 Response:
We understand that the labelling process was not clearly described, thank you for point us to this. We have added a short paragraph in the experimental results section. In short, the annotation was a manual process using the lidar data and RGB images for manually validating the objects, avoiding the need to distinguish the objects in the signal images.
1.4 Action:
We have clarified the manual labelling process at the start of the experimental results section: “We collected and manually annotated the lidar-generated signal images. We used RGB images from a separate camera and the lidar point cloud data to validate the annotations through visual observation”.
------------------------------------------------------------------------------------------------------------
Reviewer 2 Report
The manuscript analyses the performance of a variety of DL-based visual perception models in lidar camera data. It shows that general-purpose DL models can process these images to provide detection and segmentation results under indoors, outdoors, day and night environments. Comments and suggestions on the authors are as follows:
1. Please write the detailed implications of your work in the introduction. For example, what is the key benefit of this approach? What are the main problems to be solved in this paper?
2. The literature review seems to enumerate pieces of work, but the aim of a literature review is to not only give an overview of related work, but at the end of that come to a research gap, which is the gap that this paper fills.
3. The lidar camera data from the Ouster OS1-64 and the Ouster OS0-128 LiDAR are different from traditional point clouds. Of course, they are also different from RGB images. Please describe data and explain this point when revising the paper.
4. The authors showed the comparison of visualization results by multiple models which is necessary to prove the feasibility of this paper. So, I hope that the authors could use the same example data in indoor and outdoor scenarios with different models to make comparison and discussion in Figure 5.
5. Another challenge with this paper is that the effects of detection and segmentation models, are not really explained clearly. For example, the detection accuracy of YOLOv5 and Faster R-CNN for ‘Car’ is higher than YOLOx. Similar conclusions should be included in the discussion section. Although the author has not made major improvements to these methods, these conclusions are useful for researchers who use such data. Furthermore, in the abstract, please describe the detection performance of different models with numbers and % rather than words.
Author Response
************** Responses to Reviewer 2 **************
2.1 Comment:
The manuscript analyses the performance of a variety of DL-based visual perception models in lidar camera data. It shows that general-purpose DL models can process these images to provide detection and segmentation results under indoors, outdoors, day and night environments.
2.2 Response:
Thank you for your positive words and summary of our paper.
------------------------------------------------------------------------------------------------------------
2.2 Comment:
Please write the detailed implications of your work in the introduction. For example, what is the key benefit of this approach? What are the main problems to be solved in this paper?
2.2 Response:
Many thanks for your feedback, we understand that the motivation was not clear enough in the manuscript. As we mentioned to Reviewer 1, we have rephrased parts of the Introduction section to better explain the motivation between studying lidar-generated images. In short, the motivation is that lidars are already used in multiple types of autonomous mobile robots or vehicles, and we can add intelligence to the system without additional sensors (e.g., multiple cameras). Additionally, the lidar-generated images are, in general, more robust to variable weather conditions, particularly in low-light, as we show in the paper. The key benefit of the approach we propose is to get more out of the sensors that are already used.
2.2 Action:
We have rephrased the motivation across multiple paragraphs in the introduction and conclusion sections. We also added a new related work, “LiDAR-as-Camera for End-to-End Driving”. Specifically: “the authors in that paper present a novel dataset of lidar-generated images with the same lidar-as-a-camera sensor that we use in this paper. The work shows the potential of these images as they remain almost invariant across seasonal changes and environmental conditions. For example, unpaved roads can be perceived in very similar ways in summer weather, snow cover or light rain. Therefore, there is a clear advantage of these images over standard RGB or even infrared cameras, despite the limited vertical resolution”.
---------------------------------------------------------------------------------------------------------2.3 Comment:
The literature review seems to enumerate pieces of work, but the aim of a literature review is to not only give an overview of related work, but at the end of that come to a research gap, which is the gap that this paper fills.
2.3 Response:
Many thanks for your feedback, we have improved the related work section with better references to gaps in the literature in each of the subsections.
2.3 Action:
We have added closing paragraphs to each subsection in the “Related works” section. As we now explain in the paper,
in section 2.1 Lidar-based perception: “Despite the rapid advances in recent years that the works above show, processing 3D lidar point cloud data is still significantly more resource expensive than processing images. Additionally, the methods are usually purpose-built and specific to use cases or application scenarios, limiting generalizability to, for example, detecting different types of objects.”
in section 2.2 Deep learning-based object detection:” In summary, there has been an exponential increase in research in deep learning and computer vision approaches object detection. The field is significantly more mature than in lidar data processing, but there is a gap in the literature in terms of studying the applicability of these methods to other types of images generated with different sensors. In this work, we aim at studying such potential applications.”
In section 2.3. Deep Learning Based Instance and Semantic Segmentation:” Similar to object detection research, there is more mature methods in terms of computer vision for semantic segmentation compared to lidar-based segmentation. Bringing the benefits of image-based methods to lidar sensors has potential to increase the degree of situational awareness achieved across environmental conditions where passive vision sensors do no perform as well as lidars. ”
---------------------------------------------------------------------------------------------
2.4 Comment:
The lidar camera data from the Ouster OS1-64 and the Ouster OS0-128 LiDAR are different from traditional point clouds. Of course, they are also different from RGB images. Please describe data and explain this point when revising the paper.
2.4 Response:
We understand the description and characterization of data is unclear in the paper. To improve this, we have added a new reference that includes a dataset with images from Ouster lidar and more in-depth data characterization.
2.4 Action:
We have added a note in the introduction, where we first introduce sample data generated with the Ouster lidar. Since our work does not focus on characterizing the lidar data, “we refer the reader to existing dataset papers with this type of data for a more in-depth characterization of the different types of images that the Ouster lidars can generate”. The dataset paper is titled: “LiDAR-as-Camera for End-to-End Driving”
---------------------------------------------------------------------------------------------
2.5 Comment:
The authors showed the comparison of visualization results by multiple models which is necessary to prove the feasibility of this paper. So, I hope that the authors could use the same example data in indoor and outdoor scenarios with different models to make comparison and discussion in Figure 5.
2.5 Response:
Thank you for your comments. We understand that, a priori, it might seem that the same objects should be compared indoors and outdoors. Indeed, we have compared the detection of people in both cases. However, we have selected other objects that are more characteristic of the environment (e.g., chairs indoors, or cars and bikes outdoors). Importantly, as we show in our paper and is also demonstrated in other previous works (e.g., “LiDAR-as-Camera for End-to-End Driving”), one of the key benefits of lidar-generated images is that they are not affected by environmental conditions. Therefore, a person is detected in an almost invariant manner both indoors and outdoors, as long as it is at the same distance and relative position to the sensor. The same applies to daylight or nighttime lidar-generated images.
2.5 Action:
We have added a clarification in the Methodology section, with a new paragraph under the “Data Acquisition” subsection.
---------------------------------------------------------------------------------------------
2.6 Comment:
Another challenge with this paper is that the effects of detection and segmentation models, are not really explained clearly. For example, the detection accuracy of YOLOv5 and Faster R-CNN for ‘Car’ is higher than YOLOx. Similar conclusions should be included in the discussion section. Although the author has not made major improvements to these methods, these conclusions are useful for researchers who use such data. Furthermore, in the abstract, please describe the detection performance of different models with numbers and % rather than words.
2.6 Response:
Thank you for your comment. We understand that there is no clear analysis of what models perform the best. However, our objective in this paper is to study only whether the different deep learning models can adapt, without any changes to existing pre-trained neural networks, to the data generated from lidar sensors. Therefore, we believe that putting an emphasis on accuracy is counter-productive and misleading, as a proper comparison would need significantly more data.
2.6 Action:
We thank the reviewer for comments. However, we believe it is better not to emphasize further the model accuracy as we have not trained the models with new data, which would also require significantly higher amounts of labelled data. We will study this further in future works. We have added a new reference to a more recent paper where YOLOv5 is trained with lidar-generated signal images from the same Ouster sensor for the purpose of UAV tracking.
Reviewer 3 Report
1. In Related Work, it is recommended to explain the drawbacks shortly after explaining each section. This section of this article should be a review of the relevant literature, but the review process is mostly a list of literature and lack its own analysis.
2. Preprocessing is the crucial part of this article. However, the authors have mentioned only the name of the approaches. Suggesting to provide more clarity the on Data Preprocessing stage.
3. Similarly, the Image Segmentation Approaches could be explained better.
4. In line 152 stated that Object localization and detection are typically more accurate in models with a two-stage architecture than in others. If it is correct table 3 values are incorrect. In table 3 Two-Stage Object Detection models are less successful detection than One-Stage Object Detection model.
5. Why the authors do not evaluate and analyze the methods mentioned in the article from different aspects like detection speed?
6. How this approach works in unclear weather situations like fog, or rain.
7. In the abstract, the authors said that they have provided both qualitative and quantitative analysis of the performance of various neural network architectures. What is the overall performance improvement achieved by your proposed system? Provide the same in abstract as a numerical value
8. Conclusion can be accompanied by the limitation of the proposed work
Author Response
************** Responses to Reviewer 3 Starts **************
3.1 Comments:
In Related Work, it is recommended to explain the drawbacks shortly after explaining each section. This section of this article should be a review of the relevant literature, but the review process is mostly a list of literature and lack its own analysis.
3.1 Response:
Many thanks for your feedback, we have improved the related work section with better references to gaps in the literature in each of the subsections.
3.1 Action:
We have added closing paragraphs to each subsection in the “Related works” section. As we now explain in the paper,
in section 2.1 Lidar-based perception: “Despite the rapid advances in recent years that the works above show, processing 3D lidar point cloud data is still significantly more resource expensive than processing images. Additionally, the methods are usually purpose-built and specific to use cases or application scenarios, limiting generalizability to, for example, detecting different types of objects.”
in section 2.2 Deep learning-based object detection:” In summary, there has been an exponential increase in research in deep learning and computer vision approaches object detection. The field is significantly more mature than in lidar data processing, but there is a gap in the literature in terms of studying the applicability of these methods to other types of images generated with different sensors. In this work, we aim at studying such potential applications.”
In section 2.3. Deep Learning-Based Instance and Semantic Segmentation:” Similar to object detection research, there are more mature methods in terms of computer vision for semantic segmentation compared to lidar-based segmentation. Bringing the benefits of image-based methods to lidar sensors has the potential to increase the degree of situational awareness achieved across environmental conditions where passive vision sensors do not perform as well as lidars. ”
----------------------------------------
3.2 Comment:
Preprocessing is the crucial part of this article. However, the authors have mentioned only the name of the approaches. Suggesting to provide more clarity the on Data Preprocessing stage.
3.2 Response:
Thank you for your comment, we understand that the pre-processing subsection within the Methodology should be more emphasized.
3.2 Action:
We have rephrased the pre-processing subsection and added more details about the motivation as well as the specific implementation.
----------------------------------------
3.3 Comment:
Similarly, the Image Segmentation Approaches could be explained better.
3.3 Response:
Thank you for your comment, we have also addressed this issue.
3.3 Action:
We have added new details with around two pragraphs to the image segmentation subsection in Methodology section 3.5 Image Segmentation Approaches.
-----------------------------------------
3.4 Comment:
In line 152 stated that Object localization and detection are typically more accurate in models with a two-stage architecture than in others. If it is correct table 3 values are incorrect. In table 3 Two-Stage Object Detection models are less successful detection than One-Stage Object Detection model.
3.4 Response:
Thank you for your comment. It is true that single-stage models give better results for this dataset. The affirmation that two-stage detectors are better is in general and comes from the literature. In our experiments, single-stage models are indeed better, but the limited data does not allow us to draw conclusive results in this regard.
3.4 Action:
We have added a clarification to the experimental results: “In our experiments, single-stage object detectors outperform two-stage methods. The literature in the area points to the better overall performance of two-stage models. However, for the data studied in this paper, this does not hold. In any case, the limited amount of data for our tests is not enough to conclude that single-stage detectors are always better for lidar-generated data.”
--------------------------------------------------
3.5 Comment:
Why the authors do not evaluate and analyze the methods mentioned in the article from different aspects like detection speed?
3.5 Response:
Thank you for pointing this out. Several reviewers have pointed out the lack of a runtime analysis. We have added new results to the paper.
3.5 Action:
We have new evaluation results and a new subsection referring to subsection 4.3 discussing multiple detection and segmentation inference frequencies. Specifically, “we evaluated the real-time performance of multiple representatives from the above approaches including YOLOv5, Faster R-CNN from detection tasks, and PointRend from the segmentation tasks. The computing platform utilized is an Nvidia GeForce RTX 3080 GPU with 16GB GDDR6 VRAM. YOLOv5 with YOLOv5s model has an average inference frequency of 24 HZ. Faster R-CNN with ResNet50 FPN model averages to 15 HZ. Additionally, the PointRend with ResNet50 as the backbone averages 15 HZ”.
--------------------------------------------------
3.6 Comment:
How this approach works in unclear weather situations like fog, or rain.
3.6 Response:
Thank you for your comments. As we show in our paper and is also demonstrated in other previous works (e.g., “LiDAR-as-Camera for End-to-End Driving”), one of the key benefits of lidar-generated images is that they are not affected by environmental conditions. Therefore, a person is detected in an almost invariant manner both indoors and outdoors, as long as it is at the same distance and relative position to the sensor. The same applies to daylight or nighttime lidar-generated images. We have added this dataset as a new reference in our paper, since our objective is not to study the “portability” of deep learning models used for camera data in lidar-generated images.
3.6 Action:
We have added a clarification in the Methodology section, with a new paragraph under the “Data Acquisition” subsection.
-------------------------------------------------------------------------------
3.7. Comment:
In the abstract, the authors said that they have provided both qualitative and quantitative analysis of the performance of various neural network architectures. What is the overall performance improvement achieved by your proposed system? Provide the same in abstract as a numerical value.
3.7 Response:
Thank you for your comment. We understand that there is no clear analysis of what models perform the best. However, our objective in this paper is to study only whether the different deep learning models can adapt, without any changes to existing pre-trained neural networks, to the data generated from lidar sensors. Therefore, we believe that putting an emphasis on accuracy is counter-productive and misleading, as a proper comparison would need significantly more data.
3.7 Action:
We thank the reviewer for comments. However, we believe it is better not to emphasize further the model accuracy as we have not trained the models with new data, which would also require significantly higher amounts of labelled data. We will study this further in future works. We have added a new reference to a more recent paper where YOLOv5 is trained with lidar-generated signal images from the same Ouster sensor for the purpose of UAV tracking.
-------------------------------------------------------------------------------
3.8. Comment:
Conclusion can be accompanied by the limitation of the proposed work.
3.8 Response:
Thank you for your comment. We have added this to the conclusion.
3.8 Action:
We now specify limitations in the conclusion: “The main limitation of the current analysis is perhaps the lack of re-training for the models with larger datasets of lidar-generated images, owing to the lack of such annotated datasets.” We are also studying this in our next steps.
-------------------------------------------------------------------------------
************** Responses to Reviewer 3 Ends **************
Round 2
Reviewer 3 Report
All comments were resolved